# Assessment of the Elastographic and Electromyographic of Pelvic Floor Muscles in Postmenopausal Women with Stress Urinary Incontinence Symptoms

**DOI:** 10.3390/diagnostics11112051

**Published:** 2021-11-05

**Authors:** Kuba Ptaszkowski, Bartosz Małkiewicz, Romuald Zdrojowy, Malgorzata Paprocka-Borowicz, Lucyna Ptaszkowska

**Affiliations:** 1Department of Clinical Biomechanics and Physiotherapy in Motor System Disorders, Wroclaw Medical University, Grunwaldzka 2, 50-355 Wroclaw, Poland; kuba.ptaszkowski@umw.edu.pl (K.P.); malgorzata.paprocka-borowicz@umw.edu.pl (M.P.-B.); 2Department of Urology and Oncologic Urology, Wroclaw Medical University, Borowska 213, 50-556 Wroclaw, Poland; romuald.zdrojowy@umw.edu.pl; 3Institute of Health Science, University of Opole, Katowicka 68, 45-060 Opole, Poland; lucyna.ptaszkowska@uni.opole.pl

**Keywords:** elastography, stress urinary incontinence, surface electromyography, postmenopausal women

## Abstract

Background. Shear wave elastography is an effective method for studying the condition of various musculoskeletal soft tissues. The primary aim of this study was the objective elastographic and electromyographic assessment of the pelvic floor during the rest and contraction of the pelvic floor muscles (PFM) in postmenopausal women. Methods. This was a prospective observational study that was carried out at the University Hospital in Wroclaw, Poland, between January 2017 and December 2019. Patients. The target group of the study included postmenopausal women with stress urinary incontinence. The primary outcomes were the features of the elastographic assessment of the pelvic floor during rest and contraction of the PFM obtained using shear wave elastography. Results. Based on the inclusion and exclusion criteria for the study, 14 patients took part in the measurements. There was a significant difference between the elastographic assessment of the pelvic floor during rest and contraction of the PFM at all locations in front of the urethra. No statistically significant correlation was found between the results of elastography and the bioelectrical activity of PFM. Conclusion. The elasticity of the periurethral structures is higher during active pelvic floor muscle contraction than at rest, it seems that shear wave elastography is an effective test that objectively assesses the strength of PFM contraction.

## 1. Introduction

During perimenopause and early postmenopause, women experience a number of systemic and psychological changes [1,2,3]. The symptoms reported by women are classified as vasomotor and involve the urogenital system, central nervous system, musculoskeletal system and cardiovascular system [1,2,3,4]. In the present study, the most meaningful symptoms are related to the influence of the menopausal period on the pelvic floor (PF) and pelvic organs. It is believed that approximately 50% of menopausal women experience symptoms associated with the urogenital system. The most commonly reported symptoms include dyspareunia, vaginal dryness, difficulty urinating, urinary incontinence (UI), frequent urination and remittent infections [1,4,5,6,7,8].

A common cause of PF deficiency is damage to muscles, as well as connective tissue structures and nerves, as a result of numerous natural deliveries. The connective tissue has a significant function within the PF, as both ligaments and fascia are largely responsible for the stabilization of structures located in the pelvic cavity. The restoration of the correct properties of connective tissue after delivery often involves the replacement of type I collagen with the weaker type III collagen, which leads to the loss of elasticity in these structures [7,9,10,11,12,13].

The weakening of the particular components of the pelvis and PF may lead to the occurrence of pathological symptoms associated with the deterioration of their function. Therefore, a proper diagnosis and therapeutic management play a crucial role in preventing and moderating the discussed dysfunctions, especially during menopause [4,14,15,16,17,18,19].

Shear wave elastography (SWE) allows for the visualization and quantification of the stiffness of tissue in a real-time, reliable and reproducible manner. SWE is a new ultrasound-based technique for visualizing the viscoelastic properties of soft tissue. SWE is an effective method for studying the condition of various musculoskeletal soft tissues, including tendons, muscles, nerves and ligaments [20,21,22,23,24,25,26]. Previous research [20,21,22,23,24,25,26] shows the use of elastography in PFM assessment; mainly, the levator ani muscles [21,22,23,25] and the female striated urogenital sphincter [20], or the striated urethral sphincter [24], were assessed. These structures are responsible for proper urinary continence [20,24,26]. Previous scientific reports [20,24] also highlight the fact that a higher stiffness is observed during a PFM contraction, which may contribute to the stronger strain of the assessed muscles. An additional electrographic evaluation may confirm this dependence.

The primary aim of this study is the objective elastographic assessment of the PF during the rest and contraction of the pelvic floor muscles (PFM) in postmenopausal women. The main hypotheses assume that the elasticity of PF structures is higher during contraction. The secondary goal is the electromyographic assessment of PFM during contraction and relaxation and the assessment of the correlation of these results with the results of the elastographic (sEMG—surface electromyography) assessment.

## 2. Materials and Methods

### 2.1. Data Acquisition

This was a prospective observational study that was carried out at the Clinic of Urology and Urologic Oncology at the University Hospital in Wroclaw, Poland, between January 2017 and December 2019. The study protocol was approved by the Bioethics Committee of Wroclaw Medical University (KB-97/2017), and the study was conducted in accordance with the ethical principles of the Declaration of Helsinki. All patients provided written and informed consent.

Patients were recruited from the urological and gynecological outpatient clinic of the University Hospital in Wroclaw (Poland). The target group of the study included postmenopausal women. All recruited participants were screened according to the inclusion and exclusion criteria to determine their eligibility for the study. The inclusion criteria were as follows: individuals who provided informed consent to participate in the study, obtained permission from a urologist and physiotherapist to participate, did not have contraindications for the SWE measurements, and had symptoms of stress UI (SUI) (for at least 1 year) [8,27]. The exclusion criteria were as follows: a history of gynecological surgeries; a history of surgeries within the abdomen, pelvis, or lower limbs in the last 10 years; on the day of examination, the occurrence of injuries of the lower limb, pelvis or spine; organ prolapse; and third-degree UI or fecal incontinence [27]. Individuals were also excluded if they had any contraindications for SWE. 

The required sample size was estimated based on the data from a pilot study. Means and standard deviations of elastographic assessment of PF between resting activity and contraction of PFM were used in the analysis to estimate the sample size. The sample size was estimated for a two-sample, paired-means test (paired *t*-test). The estimated sample size corresponded to 14 women. The sample size was estimated using Statistica 13 (TIBCO Software Inc., Palo Alto, CA, USA).

### 2.2. Outcomes

The primary outcomes were the features of the elastographic assessment of the PF during rest and contraction of the PFM obtained using SWE. The secondary outcomes were the electromyographic assessment during rest and contraction of the PFM. The protocol of the participant examination was as follows [27]: a medical interview with a clinical assessment of SUI symptoms using the International Consultation on Incontinence Questionnaire-Urinary Incontinence Short Form (ICIQ-UI SF) was conducted; instructions on the purpose of the measurements and testing procedures were provided; consent to participate in the research was obtained; the patient was prepared for the measurements; and SWE and sEMG measurements of the PFM were taken with the patient in the supine position.

### 2.3. Ultrosound Elastography

Elastographic measurements were carried out by a SuperSonic Imagine Aixplorer^®^ apparatus (ultrasound system with Real-time ShearWave™ Elastography SWE™, Aix-en-Provence, France) with a Single Crystal Curved XC6-1 transducer. SWE was an application for real-time ultrasound that estimated stiffness (shear elastic modulus in units of kPa). The assessments were performed with patients in the supine position with both legs flexed 90 degrees at the hip/knee and an empty bladder. Before the examination, all patients were instructed on how to correctly contract the PFM. In transperineal (or translabial) ultrasound, an ultrasound probe was placed perpendicularly on the perineum [26]. The convex transducer was first positioned in the midsagittal plane and then a transducer was placed horizontally directly under the clitoris; then, the transducer was turned toward the symphysis until the bony structures became visible in the ultrasound image (puborectal-symphysis plane) [25]. The examination was conducted by an experienced urogynecologist.

In this study, imaging was performed in SWE standard mode (1.4 Hz); opacity was set to 50%; and the elasticity range was set to 150 kPa. Each area of the region of interest was marked (Q-boxes), and then the SWE image was frozen instantaneously and stored later for assessment at rest. During the contraction of the PFM, a dynamic video of the elasticity assessment was acquired. The procedure was performed 3 times for each patient and averaged. Three Q-boxes were placed posteriorly to the urethra (X, Y, Z), and three were placed anteriorly to the urethra (X, Y, Z) (Figure 1).

### 2.4. Electromyography

sEMG measurements were acquired using an MyoSystem 1400L, eight-channel electromyographic apparatus (Noraxon, Scottsdale, AZ, USA). A Life-Care Vaginal Probe PR-02 (Everyway Medical Instruments, New Taipei City, Taiwan) vaginal probe was used to record the sEMG signal from the PFM. Technical specifications: common mode rejection ratio (CMRR)—min 100 dB at 50–60 Hz; analog output gain—1000 standard (5000 selected units); input impedance >100 MΩ on sEMG channels (isolated to >3000 volts); outputs—analog +/−5 volts all sEMG channels, digital 12-bit resolution per channel from USB port; inputs—8 sEMG channels at +/−10 mV max, 8 sensor channels at +/−5 volts max, power 100–240 VAC at 50/60 Hz (0.9 A max); sEMG amplifier performance—1 µV sensitivity, <1 µV RMS baseline noise; data acquisition—12 bit resolution 8 channels, USB update to PC every millisecond; high-pass cutoff—10 Hz first order on sEMG channels; and low-pass cutoff—SelecTable 500 or 1000 Hz on sEMG channels.

The EMG signal was subjected to standard post hoc processing. First, data were rectified and smoothed using the mean square root algorithm (RMS) and then subjected to a filtering method to reduce the phase shift. A narrow band filter was used in the range from 50 to 1000 Hz (filter finite impulse response filter). The bioelectric activity of the muscles tested was expressed in microvolts (μV). The protocol of all measurements of PFM activity consisted of the assessment of the following elements: “rest tone” (10 s PFM activity at rest before functional measurements), and “contractions” (5 × 10 s contractions, in which the participant tried to contract the PFM and hold for 10 s).

### 2.5. Statistical Analysis

Statistical analysis was performed using Statistica 13 (TIBCO Software Inc.). For the measurable variables, the median and quartiles were calculated. All tested quantitative variables were tested with the Shapiro–Wilk test to determine the type of distribution. The reliability and repeatability of measurements for elastographic assessment were assessed using an intraclass correlation coefficient (ICC). In each case, r ≥ 0.90). The comparisons of results between the measurement were performed using the nonparametric U-Mann–Whitney test or chi-squared test. Correlation analysis was performed using the Spearman test. For all comparisons, the level of α = 0.05 was assumed.

## 3. Results

Thirty-four patients were eligible for the study. Based on the inclusion and exclusion criteria for the study, 14 patients took part in the measurements. The participants’ demographic and clinical characteristics are shown in Table 1.

The greatest elasticity during resting of the PFM was obtained in the Y box in front of the urethra and in the Z box from behind the urethra, and the lowest values were obtained in the X box in front of the urethra and in the X box from behind the urethra. During the contraction of PFM, the highest score was obtained in the Y box in front of the urethra and in the X box from behind the urethra. There was a significant difference between the elastographic assessment of the PF during the rest and contraction of the PFM at all locations in front of the urethra (Table 2). Statistically significant results were also observed between the sEMG results obtained at rest and during contraction (Table 3). No statistically significant correlation was found between the results of elastography and the bioelectrical activity of PFM (Table 4).

## 4. Discussion

The main aim of the study was to use SWE in the assessment of the elasticity of the PF during various PFM activities in postmenopausal women with UI symptoms. Mannella et al. [28] indicated that the main cause of urogenital symptoms in the postmenopausal period was a decrease in estrogen levels. A reduced amount of estrogen receptors was observed in the epithelium of the urinary bladder, urethra, bladder triangle, in the vaginal mucosa and in its supporting structures, i.e., the uterosacral ligament, levator ani muscle and pubo-cervical fascia [7,28]. Basha et al. [29] paid close attention to changes in the function and construction of the vaginal walls resulting from a reduced production of estrogen. Such conditions could be observed in the reduced blood flow within the vagina, decreased contractility of the vaginal smooth muscles, and changes in the construction and density of the nerve endings and collagen structures. This led to a reduction in the elasticity of the vaginal walls and, as a consequence, was conducive to the occurrence of PF dysfunction, such as organ prolapse and sexual disorders [4,7,29,30]. Petros et al. [31] emphasized the important function of the vagina in proper continence. Damages or the weakness of fascial structures supporting the anterior wall of the vagina, bladder, internal urethral orifice and urethra may lead to disorders associated with the lower urinary tract. A condition that also has an impact on the function and composition of the vagina is levator ani insufficiency (damage and/or denervation), as, through the pelvic fascia, it is combined with the smooth muscle fibers of the vagina [4,7,14,31,32].

In this study, the significant differences in the elasticity of the examined structures during the relaxation and contraction of the PFM allow us to state that this method may be effective for assessing the PF. Several research teams described elastographic assessments for the PFM [20,21,22,24,25,33,34]. The aim of the studies led by Aljuraifani and Stafford et al. [20] was to investigate the potential use of SWE as a method for assessing striated urethral sphincter contraction. Researchers assessed striated urethral sphincters in 10 healthy females using ultrasound shear wave elastography. Based on the results, they found the potential value of SWE to be a noninvasive, real-time method for assessing striated urethral sphincter function. The researchers concluded that the validation of SWE as a measuring tool for the activity of this muscle required extensive research in clinical populations, such as in females with UI. In addition, this team of researchers also assessed the use of SWE in determining the relationship between muscle stiffness and contraction intensity.

In the next study by these researchers [24], the study group was male, and the anatomical region associated with the striated urethral sphincter was assessed. The aim of the study was to check whether a higher stiffness was observed in the anatomical region associated with the striated urethral sphincter during voluntary activation. The stiffness in the area of the striated urethral sphincter was estimated via US SWE at rest and during voluntary pelvic floor muscles contractions at a 5%, 10% and 15% maximum. An EMG surface electrode was placed on the perineum to provide a general estimate of the overall intensity of the pelvic floor muscle activity. The researchers concluded that the increase in stiffness occurred in association with an increase in perineal surface electromyography activity, providing evidence that the stiffness amplitude was related to the general PFM contraction intensity [24].

It should be noted that this study also showed the difference between the elasticity of structures during rest and during the activity of these muscles in a group of people with UI. This could be important information in the context of UI diagnostics. An attempt to use SWE for diagnostic purposes was carried out by Masslo et al. [25] and Xie et al. [22]. Both teams evaluated the levator muscle. Masslo et al. [25] examined elastography as a new method for sonographic assessment of postpartum PF trauma. Based on the results, they claimed that a sonographic elastography assessment showed a postpartum trauma of the PF in women after vaginal delivery, and this new method may help identify women with a higher risk of postpartum PF disorders. On the other hand, Xie et al. [22] evaluated the variation in the stiffness of the levator in patients with stage I/II pelvic organ prolapse before and after Kegel exercises by transperineal elastography. The researchers paid attention to the effectiveness and usefulness of this method in the evaluation of the examined muscles. In consideration of our research and the literature reports, it can be concluded that elastography may have an important application in the assessment of the PF.

Our research team attempted to evaluate the structures that lay anterior and posterior to the urethra. The studies by Kreutzkamp et al. [21] emphasized the correctness of the assessments of these structures. Those researchers attempted to evaluate the correlation between the elasticity of the para-urethral tissue and urethral mobility and the correlation between the elasticity of the para-urethral tissue and UI. The researchers observed a correlation between urethral mobility and the elasticity of the para-urethral tissue; however, they found no correlation between UI and urethral elasticity.

It is interesting that no statistically significant correlations between the electromyographic and elastographic results were found, despite the literature [5,18,35] clearly demonstrating that people with weakened PFM have a lower bioelectrical activity in the electrographic assessment of these muscles. The authors of this study suggest that the lack of statistically significant correlations may result from the insufficient number of participants in the study. There is a certain tendency that shows positive correlations between these results, but the results are statistically insignificant.

The main limitation of this study is its inability to precisely determine the relevant assessed structures, which, due to individual anatomical differences, may cause problems in the repeatability of the obtained measurements, especially in follow-up assessments. An additional limitation is the lack of comparison with people who do not have a UI problem. This will be the subject of further research. Another limitation of the work is the fact that the participants had different levels of PF damage. In future studies, additional PF structure assessment methods are needed. It seems that a small number of participants should also be considered a limitation of this study.

## 5. Conclusions

In conclusion, the elasticity of the periurethral structures is higher during active PFM contraction than at rest in postmenopausal women with UI symptoms. It seems that shear wave elastography can be an effective test for objectively assessing the strength of PFM contraction. However, it is difficult to determine the diagnostic significance in UI problems, and this area of research requires further prospective studies with control groups.

## Figures and Tables

**Figure 1 diagnostics-11-02051-f001:**
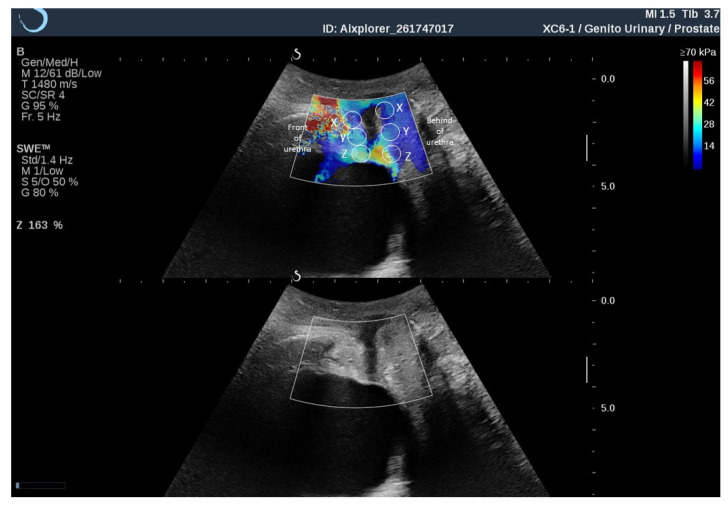
Marked places of elastography assessment (Q-boxes) of posterior to the urethra (X, Y, Z) and anteriorly to the urethra (X, Y, Z).

**Table 1 diagnostics-11-02051-t001:** Postmenopausal patients’ demographic and clinical characteristics.

Variables		Postmenopausal Women (*n* = 14)	
		Me	Q1–Q3
Age (years)		53	46–56
Weight (kg)		64	60–70
Height (m)		1.65	1.64–1.66
BMI (kg/m^2^)		24	22–24
Occurrence of urinary incontinence symptoms (years)		4	2–10
ICIQ-UI SF score		7	4–10
Qualitative variables		n	%
Kind of work	Physical	2	14
	Mental	12	86
Number of deliveries	0	1	8
	1	3	21
	2	10	71

*n*—number of participants; Me—median; Q1—first quartile; Q3—third quartile; %—percent; BMI—body mass index; ICIQ-UI SF—International Consultation on Incontinence Questionnaire-Urinary Incontinence Short Form.

**Table 2 diagnostics-11-02051-t002:** Comparison of the elastographic assessment of pelvic floor between rest tone and contraction of pelvic floor muscles.

Q-Box		Rest Tone of PFM (kPa)		Contraction of PFM (kPa)		*p*-Value *
		Me	Q1–Q3	Me	Q1–Q3	
Front of urethra	X	11.9	6.9–17.0	35.0	25.0–43.3	0.002
	Y	15.1	9.9–20.0	35.9	28.0–51.0	0.001
	Z	15.0	10.3–17.0	18.5	15.0–28.0	0.024
Behind of urethra	X	10.9	9.3–17.6	14.5	10.2–16.5	0.91
	Y	11.2	8.7–14.0	10.8	9.4–15.5	0.89
	Z	12.0	8.8–15.0	12.6	9.0–14.0	0.63

* U-Mann–Whitney test; PFM—pelvic floor muscles; kPa—kilopascal; Me—median; Q1—first quartile; Q3—third quartile.

**Table 3 diagnostics-11-02051-t003:** Comparison of the electromyographic assessment of pelvic floor between rest tone and contraction of pelvic floor muscles.

	Rest Tone of PFM (µV)		Contraction of PFM (µV)		*p*-Value *
	Me	Q1–Q3	Me	Q1–Q3	
Bioelectrical activity of PFM	3.1	2.3–3.9	9.1	8.1–11.6	<0.001

* U-Mann–Whitney test; PFM—pelvic floor muscles; µV—microvolt; Me—median; Q1—first quartile; Q3—third quartile.

**Table 4 diagnostics-11-02051-t004:** Correlation of the elastographic and electromiograhic results.

Q-Box		Rest Tone of PFM (µV)		Contraction of PFM (µV)	
		r_s_	*p*	r_s_	*p*
Front of urethra (kPa)	X	0.21	0.55	0.25	0.41
	Y	0.11	0.73	0.18	0.55
	Z	0.21	0.54	0.15	0.64
Behind of urethra (kPa)	X	0.19	0.55	0.44	0.15
	Y	0.02	0.93	0.20	0.53
	Z	0.18	0.56	0.24	0.42

PFM—pelvic floor muscles; kPa—kilopascal; µV—microvolt, r_s_—Spearman rank correlation coefficient.

## Data Availability

Data are available on request due to restrictions, e.g., privacy or ethical reasons.

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
