# Peer review of "Assessment of the Elastographic and Electromyographic of Pelvic Floor Muscles in Postmenopausal Women with Stress Urinary Incontinence Symptoms"

_diagnostics, 2021, doi:10.3390/diagnostics11112051_

Round 1

Reviewer 1 Report

It is an original study with an interesting topic.

Introduction: I would appreciate to receive more information about shear wave elastography. P.e. Previous studies with SWE and pelvic floor, why you will choose this specific technic and why it is expected that the elasticity is higher during contraction. 

2.1. line 86 exclusion criteria: All POP or POP 2? Who controlled the exclusion criteria? 

97 electromyographic (spelling)

2.3. please explain better the technic. There is also a vibration applied on the tissue. How and where is it done? A figure will be good. What does it measure exactly? Tension, strength, elasticity? Who performed the ultrasound measurement?

EMG copmarison. It would be better to do a normalization of the EMG data. This will allow to compare the EMG data.

2.5. Which ICC was applied?

Discussion:

was this technique used by other PFM SWE studies? Why you applied this technique? Why you did not measure the elevator ani muscles?

198 assessing which function of the PFM?

does it allow to assess a hyper-tone PFM?

205-207 explain it in more details

Abbreviation of PFM = pelvic floor muscles or pelvic floor muscle, use always the same.

Reviewer 2 Report

I read with great interest the manuscript, which falls within the aim of this Journal. In my honest opinion, the topic is interesting enough to attract the readers’ attention. Nevertheless, authors should clarify some points and improve the discussion, as suggested below.

Authors should consider the following recommendations:

  • Manuscript should be further revised in order to correct some typos and improve style.
  • I suggest to highlight recent data analysis about pelvic floor evaluation in female athletes (authors may refer to:  PMID: 33993850) and how the occurrence of urinary incontinence may affect sexual functions (PMID: 28439392).
